# Clinical Features of LMNA-Related Cardiomyopathy in 18 Patients and Characterization of Two Novel Variants

**DOI:** 10.3390/jcm10215075

**Published:** 2021-10-29

**Authors:** Valentina Ferradini, Joseph Cosma, Fabiana Romeo, Claudia De Masi, Michela Murdocca, Paola Spitalieri, Sara Mannucci, Giovanni Parlapiano, Francesca Di Lorenzo, Annamaria Martino, Francesco Fedele, Leonardo Calò, Giuseppe Novelli, Federica Sangiuolo, Ruggiero Mango

**Affiliations:** 1Department of Biomedicine and Prevention, University of Rome “Tor Vergata”, 00133 Rome, Italy; ferradini@med.uniroma2.it (V.F.); claudia.dem7@gmail.com (C.D.M.); miky.murdi@hotmail.it (M.M.); paola.spitalieri@uniroma2.it (P.S.); sara.mannucci@yahoo.it (S.M.); parlapiano.giovanni@gmail.com (G.P.); francescadl1992@gmail.com (F.D.L.); novelli@med.uniroma2.it (G.N.); 2Cardiology Unit, Department of Emergency and Critical Care, Tor Vergata Hospital, 00133 Rome, Italy; josephcosma1990@gmail.com (J.C.); fabiana.romeo87@gmail.com (F.R.); ruggiero.mango@gmail.com (R.M.); 3Division of Cardiology, Policlinico Casilino, 00118 Rome, Italy; martinoannamaria@yahoo.it (A.M.); leonardocalo.doc@gmail.com (L.C.); 4Department of Cardiovascular, Respiratory, Nephrology, Anesthesiology and Geriatric Sciences, Sapienza University of Rome, 00185 Rome, Italy; francesco.fedele@uniroma1.it; 5Istituto di Ricovero e Cura a Carattere Scientifico IRCCS Neuromed, Pozzilli, 86077 Isernia, Italy; 6Department of Pharmacology, School of Medicine, University of Nevada, Reno, NV 89557, USA

**Keywords:** dilated cardiomyopathy (DCM), LMNA, lamin A, lamin C, next generation sequencing (NGS)

## Abstract

Dilated cardiomyopathy (DCM) refers to a spectrum of heterogeneous myocardial disorders characterized by ventricular dilation and depressed myocardial performance in the absence of hypertension, valvular, congenital, or ischemic heart disease. Mutations in LMNA gene, encoding for lamin A/C, account for 10% of familial DCM. LMNA-related cardiomyopathies are characterized by heterogeneous clinical manifestations that vary from a predominantly structural heart disease, mainly mild-to-moderate left ventricular (LV) dilatation associated or not with conduction system abnormalities, to highly pro-arrhythmic profiles where sudden cardiac death (SCD) occurs as the first manifestation of disease in an apparently normal heart. In the present study, we select, among 77 DCM families referred to our center for genetic counselling and molecular screening, 15 patient heterozygotes for LMNA variants. Segregation analysis in the relatives evidences other eight heterozygous patients. A genotype–phenotype correlation has been performed for symptomatic subjects. Lastly, we perform in vitro functional characterization of two novel LMNA variants using dermal fibroblasts obtained from three heterozygous patients, evidencing significant differences in terms of lamin expression and nuclear morphology. Due to the high risk of SCD that characterizes patients with lamin A/C cardiomyopathy, genetic testing for LMNA gene variants is highly recommended when there is suspicion of laminopathy.

## 1. Introduction

Dilated cardiomyopathy (DCM) refers to a spectrum of heterogeneous myocardial disorders characterized by ventricular dilation and depressed myocardial performance in the absence of hypertension, valvular, congenital, or ischemic heart disease. Diverse aetiologies for DCM have been revealed, including genetic mutations, infections, inflammation, autoimmune diseases, exposure to toxins, and endocrine or neuromuscular causes [1]. As regards to genetic forms of DCM, more than 40 genes have been identified, causing defects in various cellular compartments and pathways such as the nuclear envelope, the contractile apparatus, the Z-disk, and calcium handling [2].

Mutations in LMNA (MIM 150330) gene, encoding for lamin A/C, account for 0.5–5% of DCM; however, its prevalence increases up to 10% in familial DCM and up to 33% in DCM associated to atrioventricular conduction disorders [3,4]. Lamin proteins form the nuclear lamina, a protein meshwork laying the inner surface of the nuclear envelope [5]. Lamin A and lamin C represents two isoforms encoded by a single gene (LMNA), located on chromosome 1q21.2-q21.3 [6]. Lamins, in addition to conferring cellular and nuclear integrity [7], are implicated in a plethora of crucial cellular functions, such as mechano-transduction, chromatin protection/organization, regulation of signaling, and gene expression [8,9]. To date, more than 500 LMNA variants have been reported [10] causing a wide variety of diseases and ranging from premature ageing to metabolic and skeletal muscle disorders [11,12].

LMNA-related cardiomyopathies are characterized by heterogeneous clinical manifestations that vary from a predominant structural heart disease, mainly mild-to-moderate left ventricular (LV) dilatation associated or not with conduction system abnormalities, to high pro-arrhythmic profile, where sudden cardiac death (SCD) occurs as first manifestation in an apparently normal heart [3]. Brady- and tachy-arrhythmias are a very common finding in lamin cardiomyopathies with conduction system disease commonly preceding the development of DCM by few years to a decade or more [13]. Moreover, supraventricular tachyarrhythmias (SVT) are generally more common than malignant ventricular arrhythmias (VA) at first clinical contact [14]. The mode of inheritance of cardiac laminopathies is autosomal dominant with an almost complete penetrance by the seventh decade [15,16,17]. Lamin cardiomyopathies are characterized by a poor prognosis and a high rate of major cardiac events, with the most aggressive clinical course [18].

In the present study, we report the genotype–phenotype correlation of 18 DCM patients evidenced heterozygotes for LMNA variants out of 77 referred to our Medical Genetics Unit. In three of them, functional analyses have been performed in order to validate the pathogenicity of two novel lamin variants detected during this work.

## 2. Materials and Methods

### 2.1. Study Population

Seventy-seven DCM patients and their relatives followed up at the Cardiology Unit of Policlinico Casilino (Rome, Italy) were genotyped at the Medical Genetics Unit of Tor Vergata Hospital; after genetic counselling and informed consent was signed, 15 probands evidenced heterozygotes for LMNA variants.

### 2.2. Clinical and Instrumental Characterization

Probands were defined as the first patients in a family referred for genetic testing due to a diagnosis of phenotypic DCM based on the Mestroni criteria for familial DCM [19]. The age of onset of symptoms or documented first traits of the disease was recorded. Family members who underwent genetic testing as part of family screening and had no reported cardiac symptoms at the time of the genetic testing were defined as genotype-positive phenotype-negative family members. Atrioventricular block by PR interval was assessed from a resting 12-lead electrocardiography (ECG). Arrhythmias (atrial and ventricular) were collected from a resting 12-lead ECG, exercise ECG, Holter monitoring and pacemaker, or implantable cardioverter defibrillator (ICD) monitoring. Ventricular arrhythmias were classified as non-sustained ventricular tachycardia (VT), defined as ≥3 consecutive ventricular beats with a rate ≥120/min lasting <30 s, or sustained VA, defined as VT with a rate ≥120/min lasting >30 s, ventricular fibrillation (VF), appropriate antitachycardia pacing (ATP) therapy, appropriate defibrillator shock therapy, and aborted cardiac arrest. Implantable cardioverter defibrillator and cardiac resynchronization therapy (CRT) interrogations were retrospectively reviewed and eventual therapies (ATP or defibrillator shock) recorded. Two-dimensional echocardiography was performed at the subject’s first visit using the Vivid 7 or Vivid E9 system (GE Healthcare, Horten, Norway) and analyzed using commercially available software (EchoPACVR, GE). LV ejection fraction (EF) and LV volumes were calculated from apical views using Simpson’s biplane method. Left ventricular diameters were obtained from the parasternal long-axis view. When possible, patients underwent CMR at baseline by using a 1.5-T scanner (Philips Intera CV; Philips Healthcare, Best, The Netherlands) and a phased array cardiac receiver coil, according to standard acquisition protocols set by the Society for Cardiovascular Magnetic Resonance [20]. Electrocardiogram-gated, breath-hold steady-state free precession cine images were acquired in both the long- and the short-axis planes from the LV apex to the LV base. Images were subsequently analyzed offline by using a commercially available software (View Forum software, Version 5.1, Philips Healthcare, Best, The Netherlands). LV and RV end-diastolic diameters and volumes as well as end-systolic diameters and volumes, stroke volumes, EF, and left atrium area were calculated, in accordance with the Society of Cardiac Magnetic Resonance criteria [21], by using the Extended MR WorkSpace 2.6.3.4, 2012 Philips Medical System work-station. LV dilatation was diagnosed in the presence of indexed end-diastolic volumes >81 mL/m^2^ for men and >76 mL/m^2^ for women, respectively [22].

### 2.3. Genetic Analysis

Genomic DNA was extracted from peripheral blood using EZ1 AdvancedXL (Qiagen), according to the manufacturer’s instructions. After Qubit 2.0 quantification, NGS was performed (Ion Torrent S5 and Ion Chef System) using a Custom Panel for SCD (Appendix A), designed by Ion Ampliseq Designer (Thermo Fisher Scientific, Waltham, MA, USA). Results were analyzed with Ion Reporter and Integrated Genome Viewer (IGV). The interpretation of genetic variants was conducted by Human Gene Mutation Database (HGMD), VarSome, ClinVar, Exac, and GnomAD. Moreover, DANN and Genomic Evolutionary Rate Profiling (GERP) were used. Sanger sequencing was used to confirm genetic variants and segregation analysis.

### 2.4. Fibroblasts Derivation from Skin Biopsy

Primary skin fibroblasts were obtained by a skin punch biopsy from two healthy donors (WT) and three DCM patients, after written consent. Tissues were treated as already described [23], and after 15 days, primary culture of the derived human dermal fibroblasts (HDFs) was expanded and analyzed.

### 2.5. Immunofluorescence

HDFs were incubated with primary antibodies anti-Lamin A/C (N-18, sc-6215, Santa Cruz Biotechnology) and anti-prelamin A (C-20, sc-6214, Santa Cruz Biotechnology), as described [23]. Nuclei were counterstained with HOECHST (33342, Thermo Fisher Scientific, Waltham, MA, USA). Images have been acquired by fluorescence microscope (Zeiss Axioplan).

### 2.6. Detection of Nuclear Abnormalities

For every patient’s fibroblast culture, at least 3 × 100 cells in different areas of the sample were evaluated using a Zeiss Axiplan fluorescence microscope, equipped with a 100× oil objective (Plan Apo, NA1.32). Different aspects of nuclear morphology were assessed: nuclear blebs (herniations), extensive lobulations, or donut-like invaginations of the nucleus; also, Lamin staining abnormalities were scored, including extranuclear staining and the presence of so-called honeycombs. Morphometric analysis of nuclei of HDFs WT and DCM has been performed on images from Zeiss Axiplan fluorescence microscope (Hoechst-stained nuclei), using the ImageJ processing software (http://rsbweb.nih.gov/ij/ (accessed on 20 May 2020)), by analyzing at least 10 field/sample or a minimum of 300 cells/sample. The following parameters have been analyzed by tracing nuclei and obtaining, from the ImageJ software, the following parameters: (i) nucleus area, (ii) nucleus circularity (with a value of 1.0 indicating a perfect circle), (iii) nucleus elongation (aspect ratio: the major axis over the minor axis of the fit ellipse), and (iv) nucleus roundness (the inverse of aspect ratio). The analyses have been performed on images from three different experiments, and results have been reported as mean values ± SD (fold DM vs. WT). Statistical analyses have been assessed by using the Student’s two-tailed *t*-test (* *p* < 0.05 as statistically significant).

### 2.7. Gene Expression Analysis

After TRIzol extraction (Invitrogen; Life Technologies Corporation, Carlsbad, CA, USA), total RNAs of patients and controls HDFs were DNase I (RNase-free)-treated (Ambion; Life Technologies Corporation), reverse transcribed using the High-Capacity cDNA Archive kit (Life Technologies Corporation) and used in real-time reverse transcription (RT)–polymerase chain reaction (PCR). mRNAs levels were measured by SYBR Green chemistry (Life Technologies Corporation) using the following primers: Lamin A: forward 5′-ACTGGGGAAGAAGTGGCCAT-3′; Lamin A: reverse 5′-GCTGCAGTGGGAGCCGT-3′; Lamin C: forward 5′-AACTCCACTGGGGAAGAAG-3′; Lamin C: reverse 5′-CATCTCCATCCTCATGGTC-3′; GAPDH: forward 5′-TTGCCCTCAACGACCACTTTG-3′; GAPDH: reverse 5′-CACCCTGTTGCTGTAGCCAAATTC-3′. GAPDH was used as reference gene. WT value corresponds to the mean value of two wild type samples.

### 2.8. Western Blot Assay

Proteins were extracted from patients and controls fibroblasts by RIPA Lysis buffer and Western blot analysis performed with primary antibody for Lamin A/C (N-18, sc-6215, Santa Cruz Biotechnology), followed by Mouse anti-Goat IgG (PIERCE Biotechnology). The signals were scanned and quantified on the ImageQuant LAS 4000 system, after normalizing with f β-actin. WT value corresponds to the mean value of two wild type samples.

## 3. Results

### 3.1. Lamin A/C Variants and Cardiac Phenotype among DCM Patients

Among 77 DCM families, 11 different LMNA variants were found in 15 subjects (19.5%) and confirmed by Sanger sequencing. The segregation analysis in nineteen relatives evidenced eight heterozygotes for a total of 23 (Table 1). The remaining 62 patients evidenced heterozygotes for variants in *TTN*, *DSP*, *MYBPC3*, *MYH7,* and *SCN5A* genes.

Variant classification has been made applying the ACMG/AMP guidelines [24] (Table 2). Among 23 heterozygous patients, 18 (15 males and 3 females) (Table 3) referred symptoms and/or signs of DCM, while 5 subjects were asymptomatic with apparently no signs of the disease, most likely due to their young age (from 11 to 27 years old).

All variants were located within the coil domain (Figure 1), except for W467X, a non-sense variant within the tail domain. Its clinical features are very aggressive: early onset (3rd decade), LV dilation, severe reduced systolic function, large scar in the IVS and inferior wall, and complex ventricular arrhythmias. We also characterized a proband compound heterozygous (R216H/R331L) presenting worse clinical and instrumental findings. The patient (female) with onset of symptoms in the 6th decade, and severe LV systolic dysfunction (LVEF 30%), experienced supraventricular and ventricular arrhythmias, with multiple appropriate ICD shocks.

The mean age of signs/symptoms onset in 18 symptomatic patients was 51.3 ± 12.9 years. At echocardiographic examination, mean LVEDDi was 29.2 ± 4.3 mm/m2 with LVEF 42.6 ± 10.2%, LA dilation was present in 15 of them (83.3%). Three patients (16.6%) had right heart involvement with RV dilation and dysfunction (CG11, CG12, and CG14). Cardiac MR performed in 15 phenotype-positive patients showed a LVEF of 46 ± 12% and generally a mild-to-moderate LV enlargement with mean LVEDVi 86 ± 32.8 mL/m2. A late gadolinium enhancement (LGE) as a sign of fibrosis was present in 13 of 15 affected subjects that underwent cardiac MR (86.6%), with interventricular septum (IVS) involvement in nine of them (69.2%). Analysis of basal ECG showed AV delay in 10 of 17 patients in sinus rhythm (58.8%) and IV delay in 9 of 18 patients (50%) (LBBB in 66.6%), a mean cQT of 420 ± 28 ms. The first clinical manifestation was ventricular arrhythmias (VA) in eight (44.5%) patients, advanced atrioventricular block in four (22.2%), and left ventricular dysfunction in six (33.3%). Twelve (66.6%) patients underwent ICD implantation: nine patients received ICD implantation before diagnosis of lamin cardiomyopathy was made because of the occurrence of ventricular arrhythmias in eight (CG02, CG03, CG05, CG06, CG07, CG09, CG10, and CG11) as secondary prevention and because of severe LV dysfunction in one (CG12) as primary prevention; two patients (CG8A and CG14A) whose onset was characterized by moderate LVEF dysfunction underwent ICD implantation once LMNA mutation was diagnosed, while in one patient (CG13), first clinical presentation was an AVB, and so a PMK was implanted, and, only after diagnosis of LMNA, an upgrade to ICD was performed. As regards patients CG8A, CG14A, and CG13, they received ICD implantation after genetic diagnosis of a LMNA variant according to the European Guidelines that suggest ICD implantation in lamin DCM if two of the following conditions are met: male sex, non-sustained VT, non-missense LMNA variant, and LVEF < 45% (patient CG8A: male sex, non-missense variant, LVEF 44%; patient CG14A: male sex, LVEF 45%; patient CG13: male sex, evidence of VA at PMK interrogation).

In eight patients (CG02, CG03, CG05, CG06, CG07, CG09, CG10, and CG11), whose clinical onset was characterized by ventricular arrhythmias, ICD was implanted as secondary prevention before diagnosis of lamin cardiomyopathy, as well as in another patient (CG12) that presented as the first phenotypic sign a severe reduction of the LVEF (20%); ICD was implanted as primary prevention before diagnosis of the LMNA variant.

Two patients (CG8A and CG14A) had a moderate LV dysfunction (LVEF 44% and 45%) and underwent ICD implantation after diagnosis of laminopathy because they both met the criteria for ICD implantation (as reported in the discussion, the European Guidelines recommend ICD implantation in lamin DCM if two of the following risk factors are present: male sex, non-sustained VT, non-missense variant, and LVEF < 45%):-CG08 three factors: male sex, non-missense variant, LVEF 44%-CG14A two factors: male sex, LVEF 45%

One patient (CG13) had already undergone PMK implantation for AVB, and once diagnosis of LMNA variant was made, an upgrade to ICD was performed (male sex, evidence of VA at PMK interrogation). A mean follow-up of 31.5 months was achieved, during which seven (38.8%) patients experienced AF; two patients (14%) developed new AVB. Eleven (61.1%) patients experienced VA. At ICD interrogation, 8 of 12 patients (66.6%) received an appropriate ICD therapy (shock in 62.5%) (Table 3).

One patient (CG12) received heart transplantation for end-stage heart failure. Regarding the remaining five patients without a DCM clinical phenotype, all clinical and instrumental assessments were normal except for the presence of LA dilatation in two of them (two young brothers, IV7 and IV9, belonging to Family 14, with a left atrium volume of, respectively, 36.1 and 39.6 mL/m^2^).

### 3.2. In Vitro Characterization of Two Novel LMNA Variants

One patient belonging to Family 6 (III-2) and two brothers belonging to Family 14 (IV-7 and IV-9) underwent dermal biopsy in order to obtain in vitro fibroblasts (HDFs) (Figure 2). The missense R189Q variant of unknown significance (VUS) segregates in the daughter (IV-1) but not in the sibling (III-3 and III-4) of Family 6 (Figure 2A). In Family 14 (Figure 2B), the missense E317K is classified on ClinVar as likely pathogenic for DCM and reported in an Italian family with DCM and atrioventricular block [4].

Both lamin A/C and prelamin expression have been investigated, comparing them to healthy controls (WT). Lamin A/C localization, mainly situated at the nuclear peripheric rim, was comparable between WT and DCM-derived HDFs (99.8% positive cells), after immunofluorescence staining (Figure 3A). About 4.3% of WT nuclei were positive for prelamin A, as well as the IV-7 and IV-9 nuclei (5% and 4.98%, respectively), while III-2 nuclei (Family 6) were 12.2%. In these cells, a more punctate localization pattern relative to prelamin A was also revealed. These intra-nuclear aggregates differ significantly in size and number between cells within the same culture, and they are distributed next to a typical nuclear rim (Figure 3A, III-2). The total number of abnormal nuclei, which includes herniations, honeycomb-structures, and donut-like nuclei was found to be the most discriminating parameter between patient and control cells. In fact, 6% of IV-7, 7.7% of IV-9, and 6.68% of III-2 nuclei showed an altered shape with nuclear invaginations and blebs, considered typical markers associated with LMNA variants (Figure 3A,B). Results have evidenced statistically significant differences in circularity, roundness, and nuclear elongation (Figure 3C, * *p* < 0.05). The percentage of DCM irregular nuclei strongly increases compared to WT cells, in which these alterations are present only in 1%, at the same age and number of passages (p3) (Figure 3B,C).

Successively, transcript and protein expression have been evaluated (Figure 3D), comparing quantitatively lamin A and C in DCMs with respect to WTs. In all patients, both lamin isoforms were decreased compared to WT in a statistically significant manner, except for the IV-7 patient, in whom lamin A expression slightly increased (Figure 3D). Protein quantification performed by Western blot confirmed a marked reduction of both lamin A and C isoforms (Figure 3E,F), except in IV-7, as expected, who did not show any significant protein reduction (Figure 3E).

## 4. Discussion

LMNA-related DCMs have a more aggressive clinical course compared to other forms of dilated cardiomyopathies with higher rates of potentially fatal arrhythmias and end-stage heart failure [25]. The prevalence of LMNA mutations in familial DCM is about 5–10% [26]; however, only in 30–35% of familial cases, a Mendelian inheritance has been evidenced, suggesting a prevalent complex multi-variant or oligogenic basis of inheritance [27]. Moreover, a significant clinical heterogeneity has been reported within the same family in terms of onset, severity, and progression of the disease [28]. Both the genetic and phenotypic heterogeneities together with variable penetrance of LMNA variants make the pathogenic classification of the variants difficult.

LMNA variants occur in the head and rod domains, which comprise more than half of lamin A and two thirds of lamin C, but rarely in the tail domain [28]. The clinical heterogeneity observed in LMNA-DCM might be also explained by the different functional consequences of the variants. Most carriers exhibit an age-dependent penetrance: 7% under the age of 20 years to 100% above the age of 60 years [17].

Actually, a targeted therapy is not available for the early treatment of LMNA associated cardiac disease. However, the knowledge of the genotype of DCM patients allows a better prevention with ICD because of the high risk of SCD associated with LMNA variants.

The introduction of the NGS method in the daily practice of molecular diagnosis laboratories has allowed considerably reducing response times. The possibility to identify genetic variations in DCM patients allows early diagnosis before clinical manifestation, prognosis, genetic counselling, and preventive management of heterozygous subjects and their relatives.

In our cohort of patients with DCM, LMNA variants are present in about 19.5%. This finding is not in line with previous reports describing a 5–8% prevalence [29], probably because all patients come from a clinical center specialized in arrhythmology. Affected individuals frequently suffer from a progressive conduction system disease such as atrioventricular block, bradyarrhythmias, and tachyarrhythmias and have a high chance of developing thromboembolic disorder. It has been shown that men have a worse prognosis than women [30]; however, in our study, we have only three symptomatic female patients vs. 15 male patients, and we cannot draw any conclusions about this. Three variants described in our cohort belong to Class V, seven to Class IV, and one to Class III. Interestingly, the variant E317K was reported in Gnomad with only one allele count (frequency 3.19e-5), while in our study, it is present in six proband, allowing us to hypothesize a founder effect in the Italian population, as already reported for R331Q, a founder variant in autosomal dominant cardiac laminopathy with late-onset and mild cardiac phenotypes [31]. A specific study of haplotypes associated with this variant is necessary to confirm our hypothesis. Segregation analysis performed within each family lets us identify five members in the presymptomatic stage.

In total, 11 LMNA variants were identified in 15 families, and, as expected, phenotype-positive patients showed a complex cardiac phenotype ranging from a predominantly structural heart disease to a highly arrhythmic profile in an apparently normal heart. VA were the first relief in 44.5% of patients. This finding is of particular interest since VA generally do not appear as first clinical manifestation [32], with prevalence of any and all forms of sustained VA increasing from presentation to last follow-up [33].

Moreover, 62.5% of patients who experienced VT/VF as a first manifestation had a LVEF ≥50%. At follow-up, VA occurred in up to 61.1% of patients, and 66.6% of ICD-implanted patients received appropriate ICD therapy.

These data highlight once again the highly arrhythmic burden of LMNA-related DCM and the limits of systolic function in risk stratification, emphasizing instead the importance of a gene-based diagnosis. Systolic function evaluation is an important prognostic factor in DCM, but our data suggest its marginal role in predicting the risk of VA in patients with LMNA variants. Probably the early appearance of interstitial fibrosis together with ion channels anomalies [34] could represent the cause of ventricular arrhythmias even before systolic dysfunction occurs. In fact, as already recognized [35], the presence of a scar at CMR, generally located in the IVS, is extremely frequent in asymptomatic patients, and it represents a potential trigger of VA, explaining why VA often represents the first clinical manifestation in asymptomatic patients. In our study, MRI evidenced a scar in 87.5% of the patients who experienced VA. Moreover, EF represents only one of the four independent risk factors for malignant VA identified in lamin DCM: male sex, non-sustained VT, non-missense variant, and LVEF < 45% [32]. According to the actual European Guidelines for the management of patients with ventricular arrhythmias and the prevention of sudden cardiac death, ICD implantation in patient carriers of lamin variants should be considered when two of the above mentioned criteria are met [36]. The genetic analyses would be useful not only for diagnosing the laminopathy, but also for stratifying the prognosis of carriers [37].

Patients with LMNA-related DCM frequently face supraventricular arrhythmias including atrial fibrillation, atrial flutter, and focal atrial tachycardia, as expression of atrial disease [38]. In our study, the prevalence of supraventricular arrhythmias was 38.8% at follow-up, in line with other studies [33,39]. A common relief in our population was left atrial enlargement, found in 83.3% of symptomatic patients. Left atrial size is a well-known predisposal factor for the occurrence of supraventricular arrhythmias, especially AF [40,41,42], commonly proposed as a barometer of diastolic burden and a predictor of common cardiovascular outcomes and cardiovascular death [43,44]. In our study, left atrial enlargement was the only abnormal finding evidenced in the two older male patients, aged 22 and 27 years, of the five total genotype positive asymptomatic patients. A third patient, female biological sex aged 26 years, showed an upper limit LA volume. Although this evidence comes from a low number of patients, we could speculate about atrial enlargement as an early marker of the disease in LMNA cardiomyopathies, rather than a mere consequence of pressure and/or volume overload due to LV dysfunction and worse LV compliance. The LA enlargement due to a “primitive LMNA induced atrial myopathy”, if confirmed in further studies, could represent an important relief to look for in genotype-positive phenotype-negative subjects, as initial sign of structural heart disease. Moreover, defining molecular and cellular mechanism causing the “primitive LMNA induced atrial myopathy” could lead to identify novel pathogenic mechanisms involved in supraventricular arrhythmias’ occurrence.

Successively, lamin expression and distribution were evaluated in vitro on HDFs carrying two novel LMNA variants: R189Q and E317K. Moreover, nuclear abnormalities and irregularities in lamin staining were assessed in order to correlate them to variant nature and disease phenotype [45,46,47,48,49]. The increased percentage of abnormal nuclei, irrespective of the type of nuclear malformation, is the most discriminating parameter between normal and lamin-defective cells [50], correlating with nuclear architecture instability [51]. Lamins are components of the nuclear lamina providing mechanical stability to the nucleus [52,53,54,55,56,57]. HDFs carrying R189Q displayed an abnormal prelamin accumulation, which usually correlates with senescent cells and premature aging. Accumulated prelamin A causes the captures of the transcription factor Sp1, resulting in altered extracellular matrix gene expression [58]. Nuclear dysmorphisms and nuclear envelope disorganization appear as a hallmark of human cultured laminopathic cells, independently of the associated clinical presentation [49,59,60].

Additionally, the transcription levels of both Lamin A and Lamin C have been evaluated. They are usually incorporated in the nuclear lamina in equivalent amounts and play distinct functions: any expression ratio variations may be due to altered splicing or mRNA stability [52].

In all patients, lamin C expression is statistically significantly reduced if compared to WT ones. In the III-2 patient, lamin A was also significantly reduced. Overall, lamin A/C ratio is modified from 1:1 value, especially in IV-7, in whom lamin A expression increased with respect to the younger brother. This unbalance, due to an aberrant splicing process, may lead to altered interaction with other important structural proteins and transcription factors, as well as altered chromatin interaction [61]. These data are confirmed at protein level in III-2 and IV-9 patients, expressing lower quantities of both protein isoforms, as evidenced by densitometric analysis of Western blot, suggesting a possible defective mechanotransduction and an enhanced nuclear fragility.

Importantly, the different behavior between two brothers (IV-7 and IV-9) in lamin expression can explain the worse DCM phenotype evidenced in the youngest one. These data confirm the key role played by lamin in conferring a greater susceptibility to physical stress, especially in tissues exposed to mechanical strain, such as cardiac muscle [62].

## Figures and Tables

**Figure 1 jcm-10-05075-f001:**
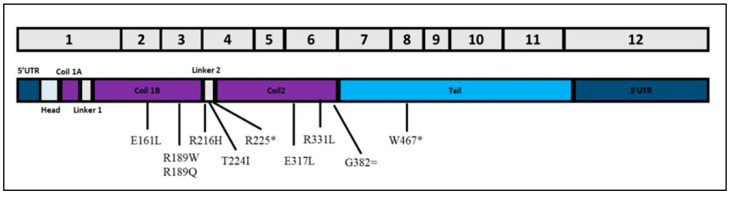
Schematic representation of lamin A transcript and localization of exonic variants identified in this study. * stands for nonsense mutations while = stands for synonymous mutations.

**Figure 2 jcm-10-05075-f002:**
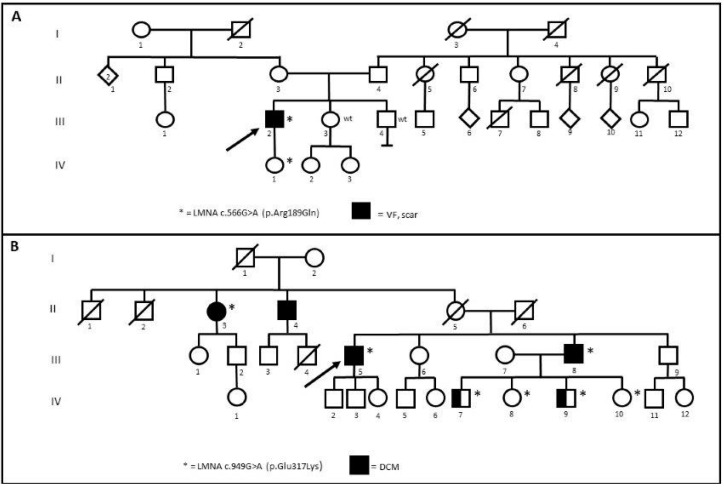
(**A**) Segregation of R189Q in Family 6. (**B**) Segregation of E317K in Family 14. Arrows indicate the proband. Wt: wild type. * stands for positive patients to Lamin A/C gene (LMNA) variants.

**Figure 3 jcm-10-05075-f003:**
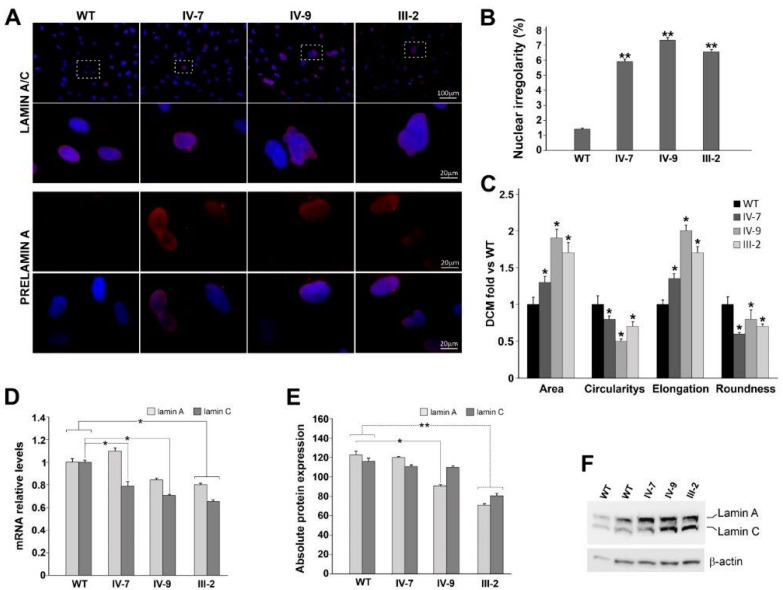
(**A**) Representative immunolabeling for lamin A/C (red) and Prelamin A (red) of WT and DCM HDFs (IV-7, IV-9, and III-2). Nuclei are counterstained with Hoechst 33,342 (blue). Scale bar 20 µm, 100 µm. (**B**) Percentage of WT and DCM cells with abnormal nuclear irregularities revealed in HDFs patients. Results represent three independent experiments with significant differences between WT and DCM HDFs (** *p* < 0.01). (**C**) Bar graphs represent the four parameters relative to nuclear shape (area; circularity; elongation; roundness); they are reported as mean values ± SD (fold DCM vs. WT) of three independent experiments. Significant differences are denoted by the *p*-value (Student’s two-tailed *t*-test; * *p* < 0.05. (**D**) RT-qPCR of lamin A and C transcripts in WT and DCM HDFs; GAPDH was used as reference gene. Data are from three independent experiments and represented as mean ± SD (* *p* < 0.05). (**E**) Densitometric analysis of Western blot performed on WT and DCM HDFs, showing the intensity of the band corresponding to lamin A and C normalized versus β-actin levels. WT densitometric value is the average between two different controls (* *p* < 0.05; ** *p* < 0.01). Data are presented as mean ± SD. (**F**) Representative Western blot of lamin A and C; β-actin is used as housekeeping gene.

**Table 1 jcm-10-05075-t001:** LMNA gene variants identified in this study.

LMNA Variant (NM_170707)	Exon	Domain	dbSNP	# Proband	# Related Individuals Carrying the Variant
E161K	2	Coil 1B	rs28933093	1 (M)	-
R189W	3	Coil 1B	rs267607626	1 (M)	-
R189Q	3	Coil 1B	rs766856162	1 (M)	1 (F)
T224I	4	Linker 2	-	1 (M)	-
R225X	4	Linker 2	rs60682848	1 (F)	-
R216H	4	Linker 2	rs757041809;/	1 (F)	-
R331L	6	Coil 2			
E317K	6	Coil 2	rs56816490	6 (5M + 1F)	5 (3M + 2F)
G382=	6	Coil 2	rs57508089	1 (M)	1 (M)
c.1381-5G > A	Intron 7		rs730880133	1 (M)	-
W467X	8	Tail	-	1 (M)	1 (M)

# stands for the number of analyzed proband for each variant on LMNA gene.

**Table 2 jcm-10-05075-t002:** Classification of LMNA variants identified.

LMNA Variant (NM_170707)	ClinVar	ACMG Classification	DANNScore	GERP	GnomAd (Allele Frequency)
E161K	Pathogenic	Likely pathogenic	0.9992	5.59	/
R189W	Uncertain significance	Likely pathogenic	0.9956	5.44	0.0000159
R189Q	Uncertain significance	Likely pathogenic	0.998	5.44	0.0000318
R216H	Uncertain significance	Likely pathogenic	0.9995	5.2699	0.0000239
T224I	/	Likely pathogenic	0.9978	5.2699	/
R225X	Pathogenic	Pathogenic	0.9974	5.2699	/
E317K	Likely pathogenic	Pathogenic	0.9992	5.67	0.0000319
R331L	/	Likely pathogenic	09987	5.67	/
G382=	Likely pathogenic/Pathogenic	Likely Pathogenic	0.7586	5.3	/
c.1381-5G > A	Uncertain significance	Uncertain significance	0.7824	5.21	0.0000482
W467X	Pathogenic	Pathogenic	0.9935	5.13	/

**Table 3 jcm-10-05075-t003:** DCM clinical phenotypes.

	CG01	CG02	CG03	CG04	CG05	CG06	CG07	CG08	CG08_A	CG09	CG10	CG10_A	CG11	CG12	CG13	CG14	CG14_A	CG15	Mean Value (n = 18)
**LMNA variant**	E317K	E317K	R189W	E317K	c.1381-5G > A	R189Q	E317K	G382=	G382=	R225X	W467X	W467X	R216H/R331L	T224I	E161K	E317K	E317K	E317K	
**Biological Sex** **(M = 1, F = 0)**	1	1	1	1	1	1	0	1	1	0	1	1	0	1	1	1	1	1	15/18 M (83.3%)
**Age at Onset (years)**	48	59	64	69	56	44	57	40	66	59	32	19	56	38	64	53	50	50	51.3 ± 12.9
**LVEDD (mm)**	65	53	52	60	64	54	50	67	57	55	57	44	54	65	51	65	51	52	56.4 ± 6.5
**LVEDDi** **(mm/m^2^)**	29.8	24.3	26.3	31.6	38.3	24.4	24.6	31.6	31.7	32.3	33.1	27	36.5	24.5	28.3	29.9	23.8	28.8	29.2 ± 4.3
**EF echo (%)**	40	55	50	37	25	51	50	35	44	53	35	50	30	20	50	45	45	51	42.6 ± 10.2
**EF CMR (%)**	31	69	60	46	20	55	55	37	/	50	40	51	39	/	/	42	45	50	46 ± 12
**LVEDV CMR (ml)**	88	99	112	247	273	151	128	243	/	158	184	138	131	/	/	172	157	150	162.1 ± 54.5
**LVEDVi CMR (mL/m^2^)**	40	45	57	130	163	68	56	115	/	87	106	85	88,5	/	/	88	79	83	86 ± 32.8
**AV delay** **(0 = no, 1 = yes)**	0	1	0	1	0	0	1	1	1	/	1	0	0	1	0	1	1	1	10/17 (58.8%)
**IV delay** **(0 = no, 1 = yes)**	1	0	0	1	0	0	0	0	1	1	0	0	1	1	1	0	1	1	9 (50%)
**LBBB** **(0 = no, 1 = yes)**	0	/	/	1	/	/	/	/	1	0	/	/	1	1	1	/	1	0	6/9 (66.6%)
**AVB** **(0 = no, 1 = yes)**	0	1	0	1	0	0	0	0	0	1	1	0	0	0	1	0	0	1	6 (33.3%)
**cQT (msec)**	430	395	408	430	428	386	400	420	425	490	400	392	470	440	390	421	438	400	420 ± 28
**LA dilatation** **(0 = no, 1 = yes)**	1	0	1	1	1	1	1	1	0	1	1	0	1	1	1	1	1	1	15 (83.3%)
**RV involvement**	0	0	0	0	0	0	0	0	0	0	0	0	1	1	0	1	0	0	3 (16.6%)
**CMR scar** **(0 = no, 1 = yes)**	1	1	1	0	1	1	0	1	/	1	1	1	1	/	/	1	1	1	13/15 (86.6%)
**Scar IVS** **involvement** **0 = no, 1 = yes)**	0	1	1	/	1	0	/	1	/	1	1	0	1	/	/	0	1	1	9/13 (69.2)
**First clinical** **manifestation** **(VA = 1, AVB = 2, LV** **dysfunction = 3)**	3	1	1	2	1	1	1	2	3	1	1	3	1	3	2	3	3	2	VA in 44.5%-AVB in 22.2%-LV dysfunction in 33.3%
**AF** **(0 = no, 1 = yes)**	1	1	1	0	0	0	0	0	0	1	1	0	0	1	0	0	0	1	7 (38.8)
**VT/VF** **(0 = no, 1 = yes)**	0	1	1	0	1	1	1	0	1	0	1	0	1	1	1	0	1	0	11 (61.1%)
**ICD** **(0 = no, 1 = yes)**	0	1	1	0	1	1	1	0	1	1	1	0	1	1	1	0	1	0	12 (66.6%)
**ICD Therapy**	/	0	1	/	0	1	1	/	1	0	1	/	1	1	0	0	1	/	8/12 (66.6)

CG: cardiogenetic samples; LMNA: Lamin A/C gene; LVEDD: left ventricular end-diastolic diameter; LVEDDi: left ventricular end-diastolic diameter index; EF: ejection fraction; CMR: cardiovascular magnetic resonance; LVEDV: left ventricular end diastolic volume; LVEDVi: left ventricular end-diastolic volume index; AV: atrioventricular delay; IV: intraventricular delay; LBBB: left bundle branch block; AVB: atrioventricular block; cQT: corrected QT interval; LA: left atrium; RV: right ventricle; IVS: interventricular septum; VA: ventricular arrhythmias; AF: atrial fibrillstion; VT/VF: ventricular tachycardia/ventricular fibrillation; ICD: implantable cardioverter–defibrillator.

## Data Availability

Variant submitted to ClinVar, Submission ID: SUB9594435.

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
