# Peer review of "Clinical Features of LMNA-Related Cardiomyopathy in 18 Patients and Characterization of Two Novel Variants"

_jcm, 2021, doi:10.3390/jcm10215075_

Round 1

Reviewer 1 Report

Review manuscript ID: jcm-1394979

Title: Clinical features of LMNA-related cardiomyopathy in 18 patients and characterization of two novel variant.

The article sent for review presents an important and current problem concerning the influence of the molecular test results in dilated cardiomyopathy on the therapeutic management and prognosis of the patients.

The etiology of DCM is very heterogeneous, therefore the possibility of performing a genetic test and analyzing the genotype-phenotype correlation is a great value of this work. It is worth emphasizing that the methodology of this study is very well presented.

A very valuable supplement to the work will be providing information in response to the questions:

  1. The Authors present 15 of the 77 subjects who were diagnosed with LMNA-related DCM. Where does the number 18 in the title of the work come from?
  2. What was the molecular aetiology of DCM in the remaining 62 patients? I suggest you mention it briefly.
  3. What were the indications for ICD implantation in the studied group of patients? I suggest that the Authors write a short summary that will be more readable and educational (apart from the data provided in the table).
  4. When was ICD implanted - right after the diagnosis of the LMNA mutation?

Reviewer 2 Report

Please change the word 'gender' with 'biological sex'. From a scientific point of view, gender doesn't mean anything. Unfortunately, many scientific papers still use the term gender but I would encourage the authors here to be more scientifically accurate.

Although, only 3 females were involved in the study, the authors should point out the biological sex of the patients in their table.

The work on patient-derived fibroblasts can be of high interest and provide some mechanistic hints. However, the data is collected from 2 wt and only 1 patient per mutation. It is hard to believe statistic with a n=1. The author should comment and use some caution on that.
